# EFFICIENT TRANSFORMERS IN REINFORCEMENT LEARNING USING ACTOR-LEARNER DISTILLATION

**Emilio Parisotto & Ruslan Salakhutdinov**
Machine Learning Department
Carnegie Mellon University
Pittsburgh, PA 15213, USA
{eparisot,rsalakhu}@cs.cmu.edu

## ABSTRACT

Many real-world applications such as robotics provide hard constraints on power and compute that limit the viable model complexity of Reinforcement Learning (RL) agents. Similarly, in many distributed RL settings, acting is done on un-accelerated hardware such as CPUs, which likewise restricts model size to prevent intractable experiment run times. These "actor-latency" constrained settings present a major obstruction to the scaling up of model complexity that has recently been extremely successful in supervised learning. To be able to utilize large model capacity while still operating within the limits imposed by the system during acting, we develop an "Actor-Learner Distillation" (ALD) procedure that leverages a continual form of distillation that transfers learning progress from a large capacity learner model to a small capacity actor model. As a case study, we develop this procedure in the context of partially-observable environments, where transformer models have had large improvements over LSTMs recently, at the cost of significantly higher computational complexity. With transformer models as the learner and LSTMs as the actor, we demonstrate in several challenging memory environments that using Actor-Learner Distillation recovers the clear sample-efficiency gains of the transformer learner model while maintaining the fast inference and reduced total training time of the LSTM actor model.

## 1 INTRODUCTION

Compared to standard supervised learning domains, reinforcement learning presents unique challenges in that the agent must act while it is learning. In certain application areas, which we term *actor-latency-constrained* settings, there exists maximum latency constraints on the acting policy which limit its model size. These constraints on latency preclude typical solutions to reducing the computational cost of high capacity models, such as model compression or off-policy reinforcement learning, as it strictly requires a low computational-complexity model to be acting during learning. Here, the major constraint is that the acting policy model must execute a single inference step within a fixed budget of time, which we denote by $T_{actor}$ – the amount of compute or resources used during learning is in contrast not highly constrained. This setting is ubiquitous within real-world application areas: for example, in the context of learning policies for robotic platforms, because of inherent limitations in compute ability due to power and weight considerations it is unlikely a large model could run fast enough to provide actions at the control frequency of the robot's motors.

However, for many of these strictly actor-latency constrained settings there are orthogonal challenges involved which prevent ease of experimentation, such as the requirement to own and maintenance real robot hardware. In order to develop a solution to actor-latency constrained settings without needing to deal with substantial externalities, we focus on the related area of distributed on-policy reinforcement learning (Mnih et al., 2016; Schulman et al., 2017; Espeholt et al., 2018). Here a central learner process receives data from a series of parallel actor processes interacting with the environment. The actor processes run step-wise policy inference to collect trajectories of interaction to provide to the learner, and they can be situated adjacent to the accelerator or distributed on different machines. With a large model capacity, the bottleneck in experiment run-times for distributed learning quickly becomes actor inference, as actors are commonly run on CPUs or devices without significant hardware acceleration available. The simplest solution to this constraint is to increase

the number of parallel actors, often resulting in excessive CPU resource usage and limiting the total number of experiments that can be run on a compute cluster. Therefore, while not a hard constraint, experiment run-time in this setting is largely dominated by actor inference speed. The distributed RL setting therefore presents itself as a accessible test-bed for solutions to actor-latency constrained reinforcement learning.

Within the domain of distributed RL, an area where reduced actor-latency during learning could make a significant impact is in the use of large Transformers (Vaswani et al., 2017) to solve partially-observable environments (Parisotto et al., 2019). Transformers (Vaswani et al., 2017) have rapidly emerged as the state-of-the-art architecture across a wide variety of sequence modeling tasks (Brown et al., 2020; Radford et al., 2019; Devlin et al., 2019) owing to their ability to arbitrarily and instantly access information across time as well as their superior scaling properties compared to recurrent architectures. Recently, their application to reinforcement learning domains has shown results surpassing previous state-of-the-art architectures while matching standard LSTMs in robustness to hyperparameter settings (Parisotto et al., 2019). However, a downside to the Transformer compared to LSTM models is its significant computational cost.

In this paper, we present a solution to actor-latency constrained settings, "Actor-Learner Distillation" (ALD), which leverages a continual form of Policy Distillation (Rusu et al., 2015; Parisotto et al., 2015) to compress, online, a larger "learner model" towards a tractable "actor model". In particular, we focus on the distributed RL setting applied to partially-observable environments, where we aim to be able to exploit the transformer model's superior *sample-efficiency* while still having parity with the LSTM model's *computational-efficiency* during acting. On challenging memory environments where the transformer has a clear advantage over the LSTM, we demonstrate our Actor-Learner Distillation procedure provides substantially improved sample efficiency while still having experiment run-time comparable to the smaller LSTM.

## 2 BACKGROUND

A Markov Decision Process (MDP) (Sutton & Barto, 1998) is a tuple of $(\mathcal{S}, \mathcal{A}, \mathcal{T}, \gamma, \mathcal{R})$ where $\mathcal{S}$ is a finite set of states, $\mathcal{A}$ is a finite action space, $\mathcal{T}(s'|s, a)$ is the transition model, $\gamma \in [0, 1]$ is a discount factor and $\mathcal{R}$ is the reward function. A stochastic policy $\pi \in \Pi$ is a mapping from states to a probability distribution over actions. The value function $V^\pi(s)$ of a policy $\pi$ for a particular state $s$ is defined as the expected future discounted return of starting at state $s$ and executing $\pi$: $V^\pi(s) = \sum_{t=0}^{\infty} \gamma^t r_t$. The optimal policy $\pi^*$ is defined as the policy with the maximum value at every state, i.e. $\forall s \in \mathcal{S}, \pi \in \Pi$ we have $V^{\pi^*}(s) > V^\pi(s)$. The optimal policy is guaranteed to exist (Sutton & Barto, 1998). In this work, we focus on Partially-Observable MDPs (POMDPs) which define environments which cannot observe the state directly and instead must reason over observations. POMDPs require the agent to reason over histories of observations in order to make an informed decision over which action to choose. This motivates the use of memory models such as LSTMs (Hochreiter & Schmidhuber, 1997) or transformers (Vaswani et al., 2017).

In this work, we use V-MPO (Song et al., 2020) as the main reinforcement learning algorithm. V-MPO, an on-policy value-based extension of the Maximum a Posteriori Policy Optimisation algorithm (Abdolmaleki et al., 2018), used an EM-style policy optimization update along with regularizing KL constraints to obtain state-of-the-art results across a wide variety of environments. Similar to Song et al. (2020), we use the IMPALA (Espeholt et al., 2018) distributed RL framework to parallelize acting and learning. Actor processes step through the environment and collect trajectories of data to send to the learner process, which batches actor trajectories and optimizes reinforcement learning objectives to update the policy parameters. The updated parameters are then communicated back to actor processes in order to maintain on-policyness.

In the following, we first refer to a single observation and associated statistics (reward, etc.) as a **step**. We refer to an **environment step** as the acquisition of a step from the environment after an action is taken. The total number of environment steps is a measure of the sample complexity of an RL algorithm. We refer to a step processed by an RL algorithm as an **agent step**. The total number of agent steps is a measure of the computational complexity of the RL algorithm. We use **steps per second** (**SPS**) to measure the speed of an algorithm. We refer to **Learner SPS** as the total number of agent steps processed by the learning algorithm per second. We refer to **Actor SPS** as the total number of environment steps acquired by a single actor per second.

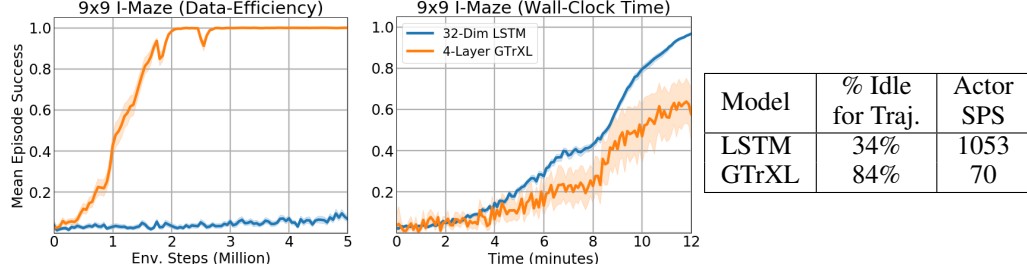

Figure 1: **Left:** On the I-Maze environment (see Sec. 5.1), there is a clear sample efficiency advantage of the Gated Transformer-XL (GTrXL) (Parisotto et al., 2019), which rapidly reaches 100% success rate compared to the LSTM. **Center:** When the x-axis is changed to wall-clock time, the LSTM becomes the more efficient model. **Right:** Table showing, for both 32-dim. LSTM and 4-layer GTrXL, (1) percentage of time the learner spends waiting for new on-policy trajectory data from distributed actor processes, (2) how many environment steps are processed per second (SPS) on a single actor process. We can see that for the much more computationally expensive transformer, the learner is spending a majority of its time idling and this is due to the order-of-magnitudes slower actor inference. Plots averaged over 3 seeds, all run using reference machine A (Appendix B).

## 3  ACTOR-LEARNER DISTILLATION

Within the setting of distributed RL, a major challenge in applying transformers to reinforcement learning is their significant computational cost owing to their high actor latency. As an example, in Fig. 1 we present some results comparing a transformer to an LSTM on the I-Maze memory environment (see Sec. 5.1). In Fig. 1, Left, a 4-layer Gated Transformer-XL (GTrXL) (Parisotto et al., 2019) agent significantly surpasses a 32-dim. LSTM in terms of data efficiency, with the x-axis as environment steps. However, in the center diagram, when the x-axis is switched to wall-clock time the LSTM becomes the clearly more efficient model.

In order to work towards an ideal model with both high sample efficiency and low experiment run time, we analyzed which parts of the distributed actor-critic system led to the transformer's main computational bottlenecks. As shown in the table on the right-hand-side of Fig. 1, we found the percentage of time that the asynchronous learner process was spending waiting for new trajectory data was substantially higher for the transformer than for the LSTM. Further looking at actor SPS between models revealed that the cause of this idling is mainly due to the transformer's actor SPS being over 15 times slower than the LSTM. This substantial decrease in SPS could even render the transformer not viable in actor-latency-constrained settings, e.g. robotic control environments, which entails a hard constraint on actor inference speed.

These findings suggest that considerable improvements in training speed could be attainable if the actors produced data faster. While de-coupling actor and learner speeds using an off-policy RL algorithm could be a suitable solution for many applications, we desired a solution that respected strict constraints on actor latency, which precludes ever running a large model during inference. As an alternative solution which respected this constraint, we designed a model compression procedure termed "Actor-Learner Distillation" (ALD) which continually performs distillation between a large-capacity "Learner" model, which is trained using RL but never run directly in the environment, and a fast "Actor" model, which is trained using distillation from the learner and is used to collect data.

As we use an actor-critic training paradigm in this work, each of these models effectively comprises two functions, the policy and value function. We denote the actor model by $\mathcal{M}_A = (\pi_A, V_A^\pi)$ and learner model by $\mathcal{M}_L = (\pi_L, V_L^\pi)$. The actor model is chosen with the appropriate model capacity to compute an inference step on the target hardware as fast as possible, i.e. within time $T_{actor}$. In contrast, there are far fewer constraints on learner model $\mathcal{M}_L$, as it is typically running on a central process with accelerated hardware. While there are no restrictions besides latency concerns on the model class of actor and learner in the context of ALD, in this work we choose to focus on the case of the actor model being an LSTM and the learner model being a transformer. As there is a clear data efficiency advantage to the transformer and a clear computational advantage to the LSTM, the success of ALD at extracting the benefits of both models can be more clearly recognized.

Actor-Learner Distillation proceeds with the learner model $\mathcal{M}_L$ being trained using a standard reinforcement learning algorithm, with the exception of the environment data being generated by the

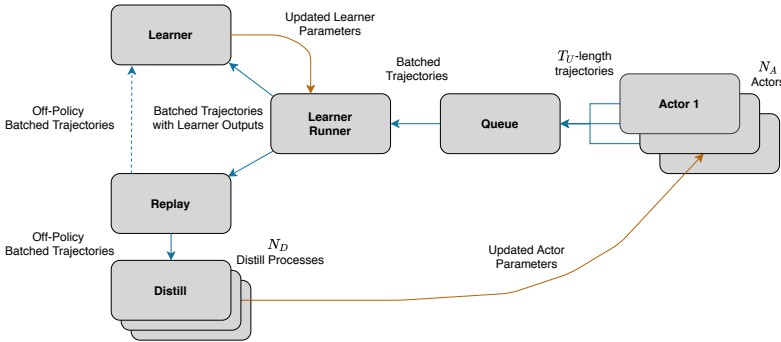

Figure 2: **Top:** An overview of distributed Actor-Learner Distillation, showing the processes as boxes and communication as arrows (data flow shown as blue arrows, parameter flow as orange).

actor model. The actor model is trained using a policy distillation loss:

$$L^\pi_{ALD} = \mathbb{E}_{s \sim \pi_A} \left[ \mathcal{D}_{KL}(\pi_A(\cdot|s)||\pi_L(\cdot|s)) \right] = \mathbb{E}_{s \sim \pi_A} \left[ \sum_{a \in \mathcal{A}} \pi_A(a|s) \log \frac{\pi_L(a|s)}{\pi_A(a|s)} \right] \quad (1)$$

Similar to previous distillation work in the multitask setting (Teh et al., 2017), we employ this loss bidirectionally: the actor is trained towards the learner policy and the learner policy is regularized towards the actor policy. This regularization loss on the learner was seen to enable smoother optimization. As actor and learner policy will naturally be different at some points during training, an off-policy RL algorithm for the learner could be thought to be required if the divergence is large enough. We found leveraging off-policy data to be effective for the actor, and sampled trajectory data from a replay buffer when computing Eq. 1 during actor model optimization.

Beyond a distillation loss, we experimented with a value distillation loss:

$$L^V_{ALD} = \mathbb{E}_{s \sim \pi_A} \left[ \frac{1}{2} (V^\pi_L(s) - V^\pi_A(s))^2 \right] \quad (2)$$

This loss functions to encourage actor model representations to model task reward structure. The use of value distillation in order to improve representations in a policy network has also been explored in concurrent work (Cobbe et al., 2020). Unlike the policy distillation loss, this loss is only used to train the actor model. The final Actor-Learner Distillation loss is:

$$L_{ALD} = \alpha_\pi L^\pi_{ALD} + \alpha_V L^V_{ALD} \quad (3)$$

where $\alpha_\pi$ and $\alpha_V$ are mixing coefficients to control the contribution of each loss.

## 4 DISTRIBUTED ACTOR-LEARNER DISTILLATION

In order to make more efficient use of computational resources, we develop a distributed system for Actor-Learner Distillation based on IMPALA (Espeholt et al., 2018). An overview of this system is shown in Fig. 2, and we detail the function of each distinct process below, in the order of data flow:

**Actor:** There are $N_A$ parallel actors, each executing on single-threaded processes with only CPU resources available. Each Actor process steps through environment steps sequentially until a trajectory of $T_U$ time steps is gathered. Actions are sampled from a local copy of the actor policy $\pi_A$. Once a completed trajectory is acquired, it is communicated to a Queue process.

**Queue:** The Queue process receives trajectories asynchronously across actors and accumulates them into batches. The batched trajectories are then passed to the Learner Runner.

**Learner Runner:** The Learner Runner process runs $\mathcal{M}_L$ in inference mode on the incoming batches of data using a hardware accelerator. This is done to provide (1) learning targets to the Distill processes downstream and (2) to make sure learner model initial memory states are updated for the Learner. Specifically when $\mathcal{M}_L$ is a transformer model, we additionally batch over the time dimension to process data even more rapidly. The Learner Runner then passes the batched trajectories it received along with computed learner model outputs to two separate parallel processing streams, the Learner process and the Replay process.

**Learner:** The Learner process receives batched trajectories from the Learner Runner and computes, using a hardware accelerator, all relevant RL objectives (V-MPO in our case (Song et al., 2020)),

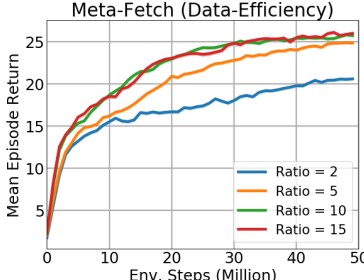 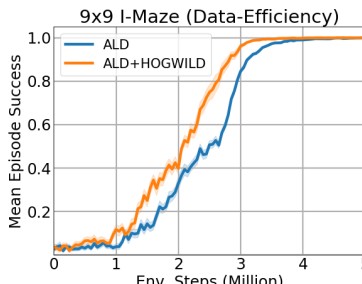

Figure 3: **Left:** On the Meta-Fetch environment with 3 objects, we evaluated the importance of the "Distillation steps per RL step" ratio. Shown on the left, as we increased this ratio we clearly observed an improved sample efficiency in the Actor-Learner Distillation procedure, until saturation at a ratio of 10. However, higher ratios came at the cost of substantial increase in wall-clock time, as the high number of distillation steps quickly becomes the bottleneck of the system's speed. **Right:** Parallelizing the distill processes using HOGWILD! improved the DpRL ratio without causing a slowdown in system speed. enabling an increase in data efficiency without a decrease in system speed, as shown in this result on 9x9 I-Maze.

along with the loss in Eq. 1 to regularize the learner model towards the actor model. In the Learner process, only the learner model parameters are updated. Similarly to recent work on deep actor-critic algorithms (Song et al., 2020; Luo et al., 2020), we use a "target network" where we only communicate updated learner parameters to the Learner Runner every $K_L$ optimization steps. Although not used here, the Learner process can additionally receive data from the Replay process.

**Replay:** The Replay process manages a replay buffer containing a large store of previously collected batched trajectories. Incoming batches of trajectories from the Learner Runner are archived in a large first-in-first-out queue. The Learner and Distill processes can then request batches of trajectories which are uniformly sampled from this queue.

**Distill:** Distill processes request data from the Replay buffer and use the retrieved trajectories to compute the distillation loss in Eq. 3, which is then used to update the actor model parameters. Similar to the Learner process, we utilize a target network scheme which updates model parameters on the Actor processes every $K_A$ optimization steps.

## 4.1 IMPROVING DISTILL / RL STEP RATIO

An observation found early in the development of ALD was the significance of the "distillation steps per RL steps" (DpRL) ratio. The DpRL ratio measures how many agent steps are taken per second on the actor model (which takes "distillation steps") comparatively with the learner model's agent steps per second (which takes "reinforcement learning steps"). This is exemplified in the left side of Fig. 3, where in the Meta-Fetch environment with 3 objects (see Sec. 5.2), we ran ALD but set a fixed number of Actor agent steps for every Learner agent step. The graph shows a greatly enhanced sample efficiency when the number of actor agent steps is increased in relation to learner agent steps, until saturating at around 10 actor agent steps per learner agent step. Out of all hyperparameters we tested, we found that a high DpRL ratio was consistently the most critical parameter in increasing the sample efficiency of ALD. However, increasing the DpRL ratio naturally increased the total run time of the system, as now the distillation process became the main system bottleneck, and this encouraged investigation into parallelized ways of improving actor agent steps.

To avoid a complete system slowdown while maintaining a high DpRL ratio, we leveraged the parallelized training procedure HOGWILD! (Recht et al., 2011) as a replacement to the single-Distill-process system originally tested. In this setting, there are now $N_D$ parallel Distill processes, all of which asynchronously sample batched trajectories from the replay buffer to update actor model parameters. As there are typically more than 1 available accelerators on a single machine (see e.g. reference machines in App. B), the parallel Distill processes can be evenly distributed between available accelerators to make the best use of the resources at hand. In the right side of Figure 3, the HOGWILD! Actor-Learner Distillation variant consistently achieved better sample complexity with equivalent time complexity, across a wide sweep of hyper-parameter settings.

## 5 EXPERIMENTS

**Algorithm Details:** For experiments, we use the V-MPO algorithm (Song et al., 2020) as the RL algorithm underlying each of the procedures we test. For ALD, we use V-MPO with V-trace correc-

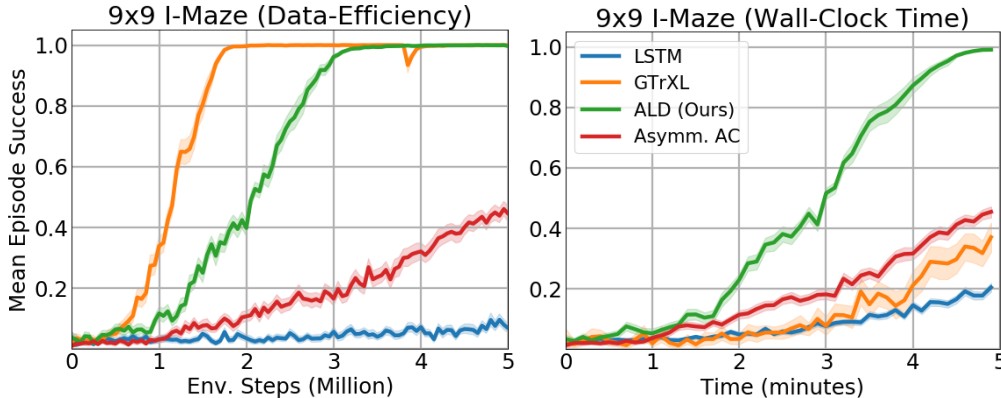

Figure 4: Results on the $9 \times 9$ I-Maze environment for all models. **Left:** x-axis as number of environment steps. **Right:** x-axis as wallclock time. All curves have 3 seeds. Obtained on reference machine A (App. B).

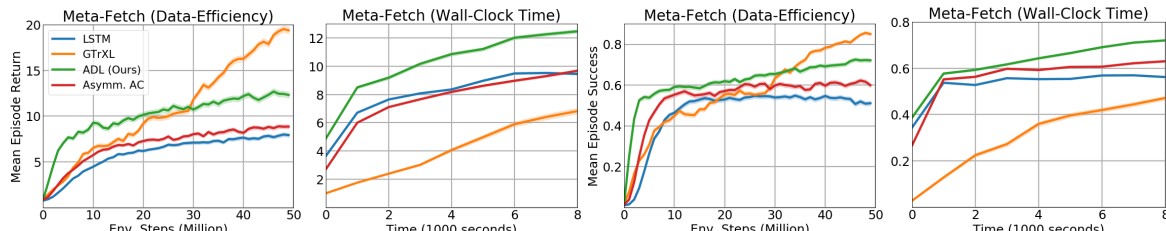

Figure 5: Results on the Meta-Fetch environment with 4 objects. **Left:** y-axis as environment returns (objects fetched correctly per episode). **Right:** y-axis as success rate (all 4 objects fetched correctly at least once in an episode). All curves have 3 seeds. Obtained on reference machine B (App. B).

tions (Espeholt et al., 2018) which worked slightly better in preliminary experiments. For baseline models V-MPO without V-trace worked better, as in Song et al. (2020). For all experiments, we use a single-layer LSTM of varying dimension depending on the environment as the actor model. For transformers, we use the Gated Transformer-XL (Parisotto et al., 2019), which we initially observed had better sample efficiency and optimization stability than standard transformers. We vary the number of transformer layers dependent on the difficulty of the environment. More details on the experimental procedure can be found in App. A.

**Baselines:** For each environment, we individually run the actor and learner model architectures used in ALD as baselines. As an additional baseline besides the learner and actor models in isolation, we introduce a model which maintains a distinct policy and value network. Here the policy network has the same architecture as the actor model in ALD (i.e. a small LSTM) and the value network has the same architecture as the learner model (i.e. a transformer). As the value function does not need to be run on actor processes, only the policy, this baseline achieves similar run time as ALD without requiring any extra insights. This baseline is an instance of Asymmetric Actor-Critic (Asymm. AC) (Pinto et al., 2017), but where instead of the value function having privileged access to state information it has privileged access to computational resources. Comparison of ALD to Asymm. AC will reveal whether the introduction of distillation losses has any positive effect over the simpler method of creating independent policy/value functions.

### 5.1 I-MAZE

The I-Maze environment has the agent start at the top-left corner of a grid-world with the shape of an "I" (Fig. 7, Left). The agent must travel from the top-left corner to one of the bottom corners of the I, where the episode is ended and the agent receives reward. The particular corner the agent must travel to is revealed in the top-right corner of the I, which contains an "indicator" tile that is either 0 or 1 depending on whether the left or right bottom corner contains the reward. If the agent does not enter a terminating corner state in $H$ time steps, the episode ends without a reward. The agent entering the correct bottom corner goal based on the indicator receives a reward of 1. Entering the incorrect corner results in the episode terminating with no reward. As memory is the main concern of these results instead of exploration, for larger maze results (i.e. a maze of width 15), we add a shaping reward of 0.01 whenever the agent visits a state along the main corridor of the "I". Once the

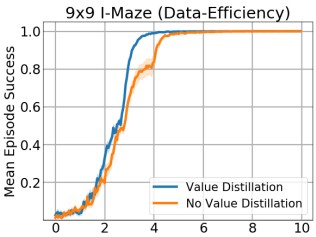 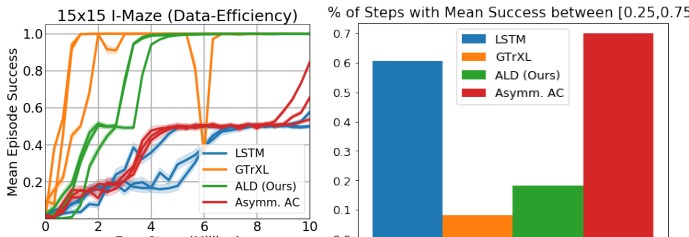

Figure 6: **Left:** Value distillation consistently provides a slight improvement to results. Results here were taken over a hyperparameter sweep with either value distillation enabled or disabled, and the best performing curves were chosen from each setting. **Center, Right:** Plotting per-seed curves (Center) of results in $15 \times 15$ I-Maze reveals that the Asymm. AC and LSTM baseline spend a significant amount of time achieving a mean success rate around 0.5 (quantified in the right-hand bar plot). A success of 0.5 suggests both models have learnt to enter a goal but are not yet able to use their memory to make the inference between goal entered and indicator identity. In contrast ALD and GTrXL rapidly learn to reach a perfect return of 1 and do not spend much time near 0.5.

agent has traversed a particular state, it cannot gain another shaping reward for returning there until the next episode. The agent has an orientation and observes every pixel directly in front of it until it reaches an occluding wall.

The environment provides a clear test on an agent's ability to remember events over long horizons, as the only way to reliably terminate an episode successfully is by remembering the indicator observed near the start of the maze. In Figure 4, we provide complete reward curves for a maze of dimension $9 \times 9$. We plot curves with two different x-axes: environment steps, measuring sample complexity, and wall-clock time. All curves were obtained on Reference Machine A (see Appendix B) under identical operating conditions, meaning wall-clock time is a fair comparison. We can first observe the clear sample efficiency gains of the transformer over the LSTM as well it's poorer performance when considering wall-clock time. Furthermore, we can observe that Actor-Learner Distillation achieves significantly improved sample complexity over the stand-alone LSTM. This confirms the ability of ALD to recover the sample-efficiency gains of the learner model. In terms of wall-clock time, the Actor-Learner Distillation procedure is unmatched by any other procedure, demonstrating its clear practical advantage.

In contrast to Actor-Learner Distillation, the Asymm. AC baseline does not seem to perform as well despite equivalent model complexity and equivalent time complexity, as it contains both identical actor and learner models. To gain insight into this result, we plotted the per-seed curves of Actor-Learner Distillation and Asymm. AC on the $15 \times 15$ I-Maze task in Figure 6. We can clearly see an underlying pattern to the seed curves, all models have a local optimum at around 0.5 success where each model class spends a varying amount of time. A return of 0.5 suggests that the model has learnt to enter a goal, but not yet learnt to use the indicator to correctly determine which goal contains the positive reward. We measured how many environment steps each model class spent in this local minima out of the 10 million total, and found that the LSTM and Asymm. AC spend a majority of their time there. While it can be expected that the transformer can easily exit this optima due to its ability to directly look back in time, interestingly ALD seems to recover the same efficiency at escaping this minima as the stand-alone transformer. This suggests that ALD can successfully impart the learner model's inductive biases to the actor model during learning in a way that is substantially more effective than just using the transformer as a value function (Asymm. AC). Finally, in the left-hand-side of Fig. 6 we ran an ablation on $9 \times 9$ I-Maze to determine whether the value distillation loss provided benefit to learning. We performed a hyperparameter sweep for each setting of value distillation enabled or disabled. The results demonstrated that using value distillation provides a slight but significant improvement.

## 5.2 META-FETCH

The "Meta-Fetch" environment, shown in Figure 7, requires an agent to fetch a number of objects distributed randomly on an empty gridworld, with each object required to be fetched in a particular hidden sequence. Crucially, in Meta-Fetch (1) the agent can not sense the unique identity of each object, meaning each object looks the same to the agent, (2) the agent does not know beforehand the sequence of objects it must obtain, and (3) observations consist of local views of pixels in front of the agent's current orientation (see Fig. 7). When the agent collects an object, it can either receive a positive reward if this object is the next correct one in the sequence, or receive no reward and

the sequence of objects is reset. To prevent cheating (i.e. the agent obtaining a reward and then immediately resetting the sequence, and repeating), the agent receives a positive reward only on the first instance of it collecting the next correct object in the sequence. Once all objects in the sequence have been fetched, the objects and rewards are reset and the environment proceeds as it did at the start of the episode (except the agent is now in the position of the last object fetched). Meta-Fetch presents a highly challenging problem for memory architectures because the agent must simultaneously learn the identity of objects by mapping their spatial relations using only local egocentric views, as well as learn a discrete search strategy to memorize the correct sequence of objects to hit. Meta-Fetch was inspired by the meta-learning environment "Number Pad" used in previous work (Humplik et al., 2019; Parisotto et al., 2019).

Results are reported in Figure 5 for a Meta-Fetch environment with 4 randomly located objects. We present plots of average return using both environment steps and wall-clock time as x-axes (left plots), along with an additional plot which measures "success" (right plots). Success in Meta-Fetch is defined as the agent completing at least 1 full object fetch sequence (agents can accomplish multiple full fetch sequences for additional reward). As in I-Maze, we re-confirm the observation that transformers achieve a clearly superior sample efficiency over LSTMs, while at the cost of a much slower wall-clock performance. Additionally, unlike I-Maze where Asymm. AC had a more positive effect on sample efficiency, there is a less substantial difference between LSTM and Asymm. AC.

However, we can see a major difference between the baselines and Actor-Learner Distillation, which early on achieves close to the transformer's sample efficiency with a much smaller model. Additionally, it seems that in this much more challenging environment, which requires both memory and compositional reasoning, the LSTM actor model in ALD suffers in performance later on when the GTrXL achieves a higher rate of increase in reward. Examining the success rate reveals an interesting trend where the ALD-trained LSTM achieves better generalization than the LSTM and Asymm. AC models. In particular, we can see that although the LSTM and Asymm. AC have average return that is increasing, their success rate is relatively stagnant (Asymm. AC) or even decreasing (LSTM). This suggests the increase in return these baselines are observing mainly stem from them learning how to search through a fraction of possible object layouts more efficiently. In contrast, the ALD-trained LSTM's success rate is correlated with its return, suggesting it is learning to succeed in a larger variety of object layouts.

## 6 RELATED WORK

Whether a deep neural network could be compressed into a simpler form was examined shortly after deep networks were shown to have widespread success (Ba & Caruana, 2014; Hinton et al., 2015). Distillation objectives revealed that not only could high-capacity teacher networks be transferred to low-capacity student networks (Ba & Caruana, 2014), but that training using these imitation objectives produced a superior low-capacity model compared to training from scratch (Hinton et al., 2015), with this effect generalizing to the case where student and teacher share equivalent capacity (Furlanello et al., 2018). Similar to this work's desired goal to leverage the rapid memorization capabilities of transformers while training an LSTM, other work showed that distillation between different architecture classes could enable the transfer of inductive biases (Kuncoro et al., 2019; Abnar et al., 2020). Within RL, distillation has most often been used as a method for stabilizing multitask learning (Rusu et al., 2015; Parisotto et al., 2015; Teh et al., 2017; Berseth et al., 2018).

The method most closely related to ALD is Mix&Match (Czarnecki et al., 2018), where an ensemble of policies is defined such that each successive model in the ensemble has increasing capacity. All models in the ensemble are distilled between each other, and a mixture of the ensemble is used for sampling trajectories, with the mixture coefficients learnt through population-based training (Jaderberg et al., 2017). There are however significant difference from ALD: the distillation procedure was meant to transfer simple, quick-learning skills from a low-capacity model to a high-capacity model (the opposite intended direction of the benefit of distillation in ALD) and a mixture policy was used to sample trajectories which did not consider actor-latency constraints (and preferred sampling from the larger model if it performed better).

Similar to ALD, various works have looked at exploiting asymmetries between acting and learning in reinforcement learning (Pinto et al., 2017; Humplik et al., 2019). In (Pinto et al., 2017), since the value function is ultimately not needed during deployment, it was given privileged access to state information during training (whereas the policy must operate directly from pixels). Our baseline

Asymm. AC was derived from this insight that value networks are solely needed during learning, but our variant had a value function that utilized privileged compute resources instead of state access during training. Another closely related method is the training paradigm of Humplik et al. (2019), where a belief network was trained to predict hidden state information in partially-observable meta-learning environments. The hidden representations from the belief network were then used as an auxiliary input to an agent network trained to solve the task, enabling representation learning to use privileged information during training.

As an alternative to model compression, architecture development can reduce the computational cost of certain model classes. Transformers in particular have had a large amount of attention towards designing more efficient alternatives (Tay et al., 2020), as gains from increased model scale have yet to saturate (Brown et al., 2020). This architectural development has taken many forms: sparsified attention (Child et al., 2019), compressed attention (Rae et al., 2020; Dai et al., 2020), use of kNN instead of soft-attention (Lample et al., 2019), etc. (See Tay et al. (2020) for a comprehensive review). However, architectural development is largely orthogonal to the ALD procedure, as it means we can further scale actor and learner models by a corresponding degree.

## 7 CONCLUSION

In conclusion, towards the development of efficient learning methods in actor-latency-constrained settings, we developed the Actor-Learner Distillation (ALD) procedure, which leverages a continual form of model compression between separate actor and learner models. In the context of learning in partially-observable environments using a distributed actor-critic system, ALD successfully demonstrated the ability to largely capture the sample efficiency gains of a larger transformer learner model while still maintaining the reduced computational cost and lower experiment run time of the LSTM actor model. As supervised learning demonstrates ever increasing gains in performance from increasing model capacity, we believe the development of effective methods of model compression for RL will become a more prominent area of study in the future. In future work, we wish to investigate whether integrating the ALD procedure with batched inference for actors would still maintain the same performance increases we demonstrated in our results, while at the same time enabling larger actor models to be used and correspondingly larger learners.

### ACKNOWLEDGMENTS

This work was supported in part by the NSF IIS1763562 and ONR Grant N000141812861. We would also like to acknowledge NVIDIA's GPU support.

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

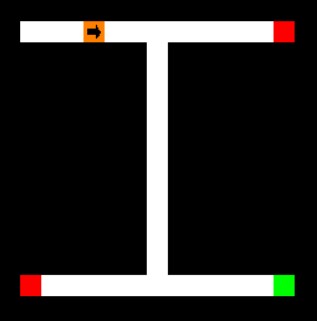 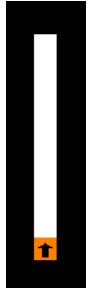 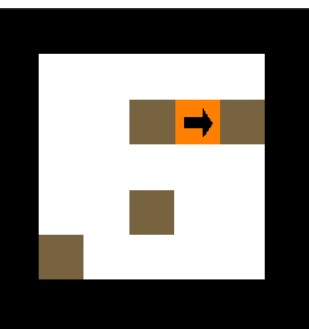

Figure 7: **Left:** Example of the $15 \times 15$-size I-Maze environment, with the indicator in this case being red. **Center:** An example of the agent observation, which is a 3-pixel-wide egocentric beam starting in the direction of the agent's orientation. The agent cannot see behind itself or through walls. **Right:** Example of the Meta-Fetch environment with 4 randomly located objects shown in brown. The agent is shown in orange with its orientation indicated by the direction of the arrow.

## APPENDIX

## A  EXPERIMENT DETAILS

For all models, we sweep over the V-MPO target network update frequency $K_L \in \{1, 10, 100\}$. In initial experiments, we also swept the "Initial $\alpha$" setting over values $\{0.1, 0.5, 1.0, 5.0\}$. All experiment runs have 3 unique seeds. For each model, we choose the hyperparameter setting that achieved highest mean return over all seeds. Additionally we use PopArt (Hessel et al., 2019) for the value output.

| Hyperparameter | Value |
|---|---|
| Optimizer | Adam |
| Learning Rate | 0.0001 |
| $N_A$ | 30 |
| $N_D$ | 8 |
| Batch Size | 64 |
| $T_U$ | 20 |
| Discount Factor ($\gamma$) | 0.99 |
| Grad. Norm. Clipping | Disabled |
| Initial $\eta$ | 1.0 |
| Initial $\alpha$ | $\{0.1, 0.5, 1.0, 5.0\}$ |
| $\epsilon_\eta$ | 0.1 |
| $\epsilon_\alpha$ | 0.004 |
| $K_L$ | $\{1, 10, 100\}$ |
| PopArt Decay | 0.0003 |

Table 1: Common hyperparameters across experiments.

### A.1  I-MAZE EXPERIMENTS

**Observations, Actions and Metrics:** Observations in I-Maze are the single row of pixels starting at the agent's current position and extending in the direction the agent is facing. The row of pixels extends as far as the maze dimension. If the agent has a wall in its field of view (represented as black pixels in Fig. 7), pixels further away from that point are occluded. For the $9 \times 9$ maze, a reward is only given when the agent enters the correct goal as decided by the indicator. For the $15 \times 15$ maze, it is the same but there is an additional shaping reward where whenever the agent enters one of the states along the central column of the "I" for the first time within its current episode, it receives a small reward of 0.01. This reward is meant to encourage exploration and prevent the very large number of environment steps otherwise necessary for the agent to end up at a goal when

taking actions randomly. The agent has 4 actions, move forward, turn left, turn right and do nothing. Moving forward into a wall causes no change in state. $9 \times 9$ I-Maze episodes last for a horizon of 150 time steps while $15 \times 15$ I-Maze episodes last for 350 steps. Observations have 3 channel dimensions, one for whether the pixel is free space or a wall, one for the green goal indicator and one for the red goal indicator.

**LSTM:** We use a single-layer LSTM with a hidden dimension of 32 for all experiments in this domain.

**GTrXL:** We use a 2-layer GTrXL for the experiments in $9 \times 9$ and a 4-layer GTrXL for the experiments in $9 \times 9$. We use an embedding size of 256, 8 attention heads, a head dimension of 32, a gated initialization bias of 2 and a memory length of 64. Other details were followed from Parisotto et al. (2019).

**ALD:** We use the corresponding LSTMs and GTrXL described above depending on the environment. During hyperparameter sweeps, we tested $K_A \in \{1, 10, 100\}$. We set $\alpha_\pi = 1$ and sweep $\alpha_V \in \{0, 0.1, 1\}$.

**Asymm. AC:** We use the corresponding LSTMs and GTrXL described above depending on the environment.

## A.2 META-FETCH

**Observations, Actions and Metrics:** In Meta-Fetch, an agent acts in an empty 2D gridworld of size $7 \times 7$ given only local observations. At each step the agent can choose to either move forward or turn left or right. Observations are the 15 pixels in front of the agent pointing in it's orientation, along with the 2 pixels on either side of this ray, for a complete observation size of $C \times 15 \times 3$ pixels (see Fig. 7). Each observation has $C$ channels which determine whether that pixel represents free space, a wall, an object, or a collected object. Every episode has the agent located in a new configuration of object locations. We set a maximum episode length of 300 steps for Meta-Fetch.

The goal of the agent in Meta-Fetch is to collect, by colliding with, each of the randomly located objects in a specific sequence. This sequence is not known beforehand and is randomized each episode. The only clue to discovering this sequence is that when the agent hits the next right object in the sequence, it gains a reward of 1 and the object is sensed as "collected". Otherwise, if it hit the wrong object in the sequence, then all previously collected objects are reset and the agent must restart the fetch sequence from the beginning once again. Once all objects are collected, they are all reset and the agent can restart the same sequence to once again gain reward for each collected reward. Objects which have been reset before the complete fetch sequence is achieved do not give further reward upon re-collection, to prevent the agent from falling into an local optimum of repeatedly collecting a correct object and then immediately collecting an incorrect object to reset the correct object.

**LSTM:** We use a single-layer LSTM with a hidden dimension of 128.

**GTrXL:** We use a 4-layer GTrXL, an embedding size of 256, 8 attention heads, a head dimension of 32, a gated initialization bias of 2 and a memory length of 64. Other details were followed from Parisotto et al. (2019).

**ALD:** We use the corresponding LSTMs and GTrXL described above depending on the environment. During hyperparameter sweeps, we tested $K_A \in \{10, 100\}$. We set $\alpha_\pi = 1$ and sweep $\alpha_V \in \{0, 0.1, 1\}$.

**Asymm. AC:** We use the corresponding LSTMs and GTrXL described above depending on the environment.

## B COMPUTE DETAILS

**Reference Machine A:** Reference Machine A has a 36-thread Intel(R) Core(TM) i9-7980XE CPU @ 2.60GHz, 64GB of RAM, and 2 GPUs: a GeForce GTX 1080 Ti and a TITAN V.

**Reference Machine B:** Reference Machine B has a 40-thread Intel(R) Xeon(R) CPU E5-2630 v4 @ 2.20GHz, 256GB of RAM and 2 GPUs: a Tesla P40 and a Tesla V100-PCIE-16GB.

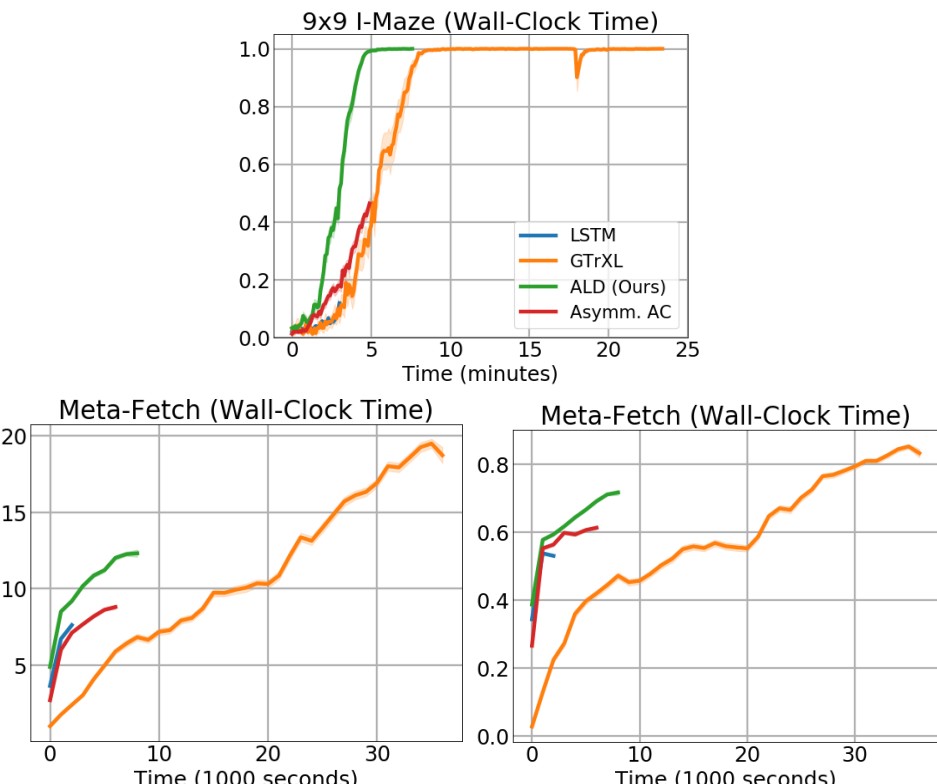

Figure 8: Full wall-clock time curves. **Top:** 9x9 I-Maze success rate. **Left:** Meta-Fetch reward. **Right:** Meta-Fetch success.

## C    DESCRIPTION OF HOGWILD!

In order to speed-up certain learning aspects of our proposed method, we use the HOGWILD! (Recht et al., 2011) algorithm. HOGWILD! enables lock-free stochastic gradient optimization to be performed by several parallel learning processes. Each process has shared access to the parameter vector of the learning model, and can update it without requiring a mutually-exclusive lock. This allows parameter updates to occur in parallel as soon as they are available, but potentially leads to race conditions where the parameter vector is read during it being updated by a different process. This means a fraction of the parameter vector can be stale when a learning process calculates the parameter gradients. However, while this parallel overwriting of the parameter vector introduces noise via this staleness, it usually does not significantly inhibit performance and is typically faster than alternatives which use explicit locking to prevent race conditions.

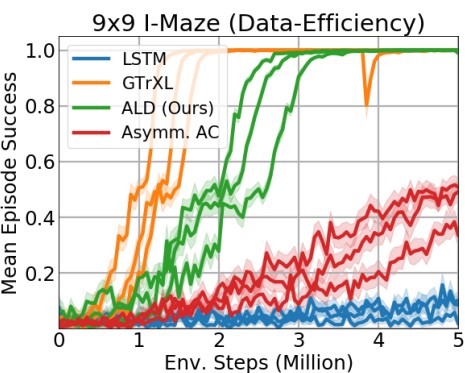

Figure 9: Per-seed curves for 9x9 I-Maze.

