# OpenReview forum: "Efficient Transformers in Reinforcement Learning using Actor-Learner Distillation"
_ICLR.cc/2021/Conference — ICLR 2021 Poster_

### Official Review · AnonReviewer1 · 2020-10-17
**Interesting idea, more visually complex environments would strengthen the paper**

**Rating:** 5
**Confidence:** 4

**Review:**

Summary:

The paper proposes a method for "actor-latency constrained" settings: Recently, transformers have been shown to be powerful models in RL which, in particular, exhibited better sample complexity in settings in which long-term credit assignment in partial observability was required (e.g. the T-maze). However, they are computationally expensive. Consequently, the authors propose to train transformers on the learner, supported by hardware acceleration, but also train a smaller LSTM agent which can be efficiently executed on the actors.

Positive points:
(+) I think this is a relevant problem setting
(+) Clear description of the algorithm which seems well thought-out

Possible weakness:
(-) Experimental evaluation.

I'm currently recommending a weak rejection. I believe the idea and execution of the paper (in particular the method part) is good. However, a stronger experimental section which also evaluates on more complex visual domains would greatly strengthen the paper. Doing so on tasks with an equal complexity on the required long-term memory might be computationally challenging, however, I personally would already be satisfied with showing that the proposed algorithm doesn't perform worse than the baselines, even on tasks which don't require (as much) memory.

In other words: The current experiments clearly show why and when ALD can provide an advantage. What is missing for me is a visually more complex experiment that shows me that the actor/learner split and associated off-policy-ness doesn't create additional problems on (visually) more complex environments. If those environments don't require as much memory, there is no reason to expect that ALD can outperform the baseline, which would be ok for me, as long as it doesn't underperform it.

One additional question (unrelated to evaluation of the paper) I was wondering: The authors mention Teh et al. What I am wondering is why, when updating pi_A, you only use the distillation loss and not also the RL loss? Did you try?

Other minor point (no impact on evaluation): I would have found a brief description of HOGWILD, e.g. in the appendix, to be helpful.

---

> ### Author Response · Authors · 2020-11-18
> **Re: Interesting idea, more visually complex environments would strengthen the paper**
>
> Dear Reviewer,
>
> Thank you for your review.
>
> Regarding the concern about the extent of our experimental evaluation, we did make an attempt to recreate a result of the gated transformer applied to DMLab, as was done in (Parisotto et al., 2019), in particular the arbitrary_visuomotor_mapping environment which included image observations and significant partial-observability. Unfortunately there was difficulty in getting a working model at the compute scale we had available. One problem was that, at the smaller transformer size (4 layers w/ 64 memlen in our case v.s. 12 layers w/ 512 memlen for the original result), the CNN to process the 96x72x3 image became a more significant bottleneck on CPU. Additionally, with the 4 layer+64 mem gated transformer and a smaller CNN we were not able to solve psychlab_arbitrary_visuomotor_mapping in the 1B env steps the authors reported for their model in the paper. The smaller gated transformer baseline model took over 16 days to train to 1B steps and we were therefore unable to experiment with a varied set of hyperparameters in the interim since submitting.
>
>
> **When updating pi_A, you only use the distillation loss and not also the RL loss? Did you try?**
> We did not try to combine the RL loss with the distillation loss on the actor model, but it is definitely a good idea and might improve ALD performance further.
>
> **Brief description of HOGWILD in the appendix**
> Thank you for suggesting this addition, we will include a summary in the updated revision’s appendix.

---

> > ### Comment · AnonReviewer1 · 2020-11-18
> > **Thank you for your reply**
> >
> > Dear Authors,
> >
> > Thank you for your reply and the paper updates!
> >
> > I've got two follow up questions:
> >
> > a) I understand that CNNs on such visually complex environments are hard on CPUs and can greatly slow down learning. However, would it be possible (in principle - not necessarily with your current architecture) to run an ablation study "centralized", i.e. to use the GPU also for inference during rollouts by batching the environment steps? While this is of course not the aim of your work, could this serve as an ablation study to show that the off-policyness of using the distilled actor is not a problem, even on visually demanding environments?
> >
> > b)  Alternatively, is there a good argument (which I am missing) that such an off-policyness should not be a problem?

---

> > > ### Author Response · Authors · 2020-11-18
> > > **Re: Thank you for your reply**
> > >
> > > Dear Reviewer,
> > >
> > > We agree that your concern about potential off-policyness is well-founded. We want to make the argument that if our currently proposed ALD was not robust to a more significant off-policyness (perhaps caused by e.g. CNN state encodings), we could extend ALD to use an off-policy RL algorithm in place of V-MPO to train the learner (such as e.g. MPO). In that case, the RL algorithm would be robust to a more substantial divergence between actor and learner than it might otherwise be with the more on-policy V-MPO + V-trace corrections algorithm.

---

### Official Review · AnonReviewer2 · 2020-10-26

**Rating:** 7
**Confidence:** 4

**Review:**

### Summary

The paper proposes an original idea to use distillation to speed up modern distributed RL settings, when data collection is done on CPUs with the learning happening on accelerated hardware, e.g. GPU. More specifically, the authors propose to use a transformer for the learner and distil the policy into LSTM actors. With this, they achieve much faster wallclock time compared to the Transformer for Actors setup, however, losing in sample-efficiency.

### Pros

 - The paper tackles a practical problem arising when designing an RL learning pipeline.
 - The paper proposes an original idea of applying distillation in the RL setting to solve a problem above.
 - Implicitly (because there is no discussion of this), the paper raises an important question of a trade-off between sample efficiency and computational efficiency in RL.

### Cons

- Confusing positioning of the paper;
- Implicit assumptions not discussed/challenged in the paper;
- Confusing results presentation (plotting, result interpretation);

### Reasoning behind the score

I believe this paper has great potential and can be improved in two possible ways. First, which will make it more general and increase its impact, is to turn it into a discussion of the trade-off between sample efficiency and computation complexity. Second, to narrow the scope and improve the positioning/clarity to get an improved version of this submission. I would really like the authors to choose the first route, but it is their decision.  At this moment, due to the pros and cons I described above, I give it a 6, Marginally Above Acceptance Threshold.

### Questions to the authors

- The paper assumes that it's better to lose in sample efficiency while computing faster. Why do you think it should be like this? What is the motivation behind such an assumption? Can you discuss the sample/computation efficiency trade-off in the paper?
- The paper assumes that LSTMs are less computationally efficient than Transformers. What exactly do you mean by computational efficiency? Yes, your plots show that the particular model instances make LSTM to achieve better performance with smaller wall-clock time. But is it general? One of the reasons behind transformers' success is their ability to parallelise. Can you explicitly state the assumptions for which you are solving the posed problem?
- The storyline in the abstract and in the intro/conclusion do not quite fit. For example, the abstract goes from distributed RL setting → constrained actors → cannot train more complex models. But in the intro, you start with the 'Transformers are great' story. In addition, in the second paragraph of the intro, you mention 'actor-latency-constrained' settings when 'there exists a max latency constraint on the acting policy'. I don't think you consider this setting later in your experiments or before (in the abstract). What is the exact problem you are solving?
- "An agent designer has to make a hard decision between ... better sample efficiency ... or ... lower wall-clock time and reduced computational cost". Can you elaborate a bit more on the setting? Ideally, describe this trade-off in more detail. Sometimes, sample-efficiency and time are intertwined, if your data collection is slow, then sample inefficiency will bump the wall-clock time.
- I'm really confused by the presentation of the results. Can you, please, plot the curves till convergence of all the algorithms? For example, the left subplot in Figure 1 shows the transformer converged to 1.0, but not the LSTM. At the same time, the next subplot shows the LSTM converged, but not the transformer. I think this is of utmost importance for better interpretation of the results. Same holds for other figures.
- For the table on top of the page 3, how do you get the numbers? At which point in training did you do the measurements? Did you average?
- On page 6 you say "... we can observe that ALD achieves near sample complexity parity to the transformer reaching a success rate of 1 in less than twice as much steps." Why is it a near parity (2X!)?
- Intro says 'ALD provides sample efficiency on par with transformers'. This is not what the plots show (e.g. 2x difference on Fig 4, 12.5 vs 20 score on Fig 5 and 0.7 vs >0.8 in episode success on the 3rd subplot of Fig 5. What is your definition of 'to be on par'?

### Other comments

- Background
    - You cite Puterman for MDPs, however, Puterman does not include the discounting coefficient into the MDP tuple.
    - Can you give a page of Sutton&Barto for your reference for the existence of an optimal policy?
    - You say you focus on POMDPs, but later use $s$ as inputs to the policy. This feels weird.
    - End of the first paragraph of the background: 'recurrent LSTMs'. Are there non-recurrent LSTMs?
    - Great second paragraph describing the glossary, I really like it.
- Distributed Actor-Learner Distillation
    - I think, V-MPO comes too late here. I've had the question about the learning algorithm several times, before I found it here. The same hold for IMPALA. I think this should be mentioned in the introduction.
    - I like the description of the algorithms in the paragraph, however, I would love to have pseudocode in the appendix.
    - "which are uniformly sampled from the FIFO queue": How does FIFO affect sampling?
- Experiments
    - "For each environment, we run the actor and learner model architectures used in ALD as baselines, where each of these models are independently trained using standard RL". This sentence is a bit confusing. Also, can you explain what does 'standard RL mean?
    - "Comparison of ALD to Asymm." should be on the next line, I believe.
    - Why did you decide to do you present the per-seed curves on 15x15 maze while presenting the aggregated results for 9x9? How do these look on 9x9 maze?
- Related Work
    - It would be great to have a paragraph giving pointers to the existing distributed RL frameworks and some questions the papers introducing them investigate.

---

> ### Author Response · Authors · 2020-11-18
> **Re: Review Part 1**
>
> Dear Reviewer,
>
> Thank you for your review and in-depth feedback. We hope to address your questions below:
>
> **The paper assumes that it's better to lose in sample efficiency while computing faster. Why do you think it should be like this? What is the motivation behind such an assumption? Can you discuss the sample/computation efficiency trade-off in the paper?**
>
> We do not necessarily make an assumption that decreased total wall-clock time is better than sample efficiency. In our setting, we want to consider the case where there is a hard constraint on model size that can be used at deployment (i.e. acting/inference), for example it is running on-board a real robot and therefore has limits on weight and power consumption. In this setting, you would either have to train the low-complexity model directly or apply our ALD procedure. Our results consistently demonstrated a significant and unambiguous improvement over the low-complexity model, which is technically the main result of our paper. We use the proxy of distributed RL, which is a setting without strict latency constraints, to demonstrate the effectiveness of our procedure due to its convenience and ease of experimentation.
> We believe a thorough analysis on the trade-off between sample and compute efficiency is currently beyond the scope of our paper. Quantifying this trade-off in general seems quite difficult, and we currently do not have means to do this in the paper beyond the empirical results obtained on the memory tasks we tested.
>
> **The paper assumes that LSTMs are less computationally efficient than Transformers. What exactly do you mean by computational efficiency? ... is it general? One of the reasons behind transformers' success is their ability to parallelise. Can you explicitly state the assumptions for which you are solving the posed problem?**
>
> In the paper, we measure two axes of model performance: sample efficiency and wall-clock time. Our experiments revealed that transformers are more sample efficient than LSTMs (this has also been previously reported in the literature for RL in Parisotto et al., 2019), but in terms of wall-clock time LSTMs can sometimes achieve the same performance, at least in the environments we tested. We think determining whether this is general across all settings is beyond the scope of this paper, but we want to emphasize that the ALD procedure is not intrinsically tied to transformers and LSTMs. For example, there is nothing blocking the application of ALD with large LSTM learner and small LSTM actor. We chose the particular setting of transformers and LSTMs since the performance characteristics of ALD over baselines are clear. We will clarify this in the introduction.
>
>
> **The storyline in the abstract and in the intro/conclusion do not quite fit. What is the exact problem you are solving?**
>
> Thank you for the feedback on the storyline, we will improve and clarify this in the next revision. To summarize here, the exact problem we want to solve is the “actor-latency-constrained” setting, where acting has an upper limit on policy model complexity. These settings are important but there are additional complexities such as limited number of robot platforms available for generating data, the chance of real robot damage, currently limited capabilities (memoryless tasks such as locomotion are already very challenging), etc. To avoid these additional complexities, and to have a setting where we can more easily ablate design considerations (like the DpRL ratio), we chose the proxy of distributed RL. Distributed RL is not a strictly latency constrained setting -- you can technically inference any model on CPU, it just might take an extremely long time. While latency is not a strict constraint in distributed RL, it is a significant practical consideration as faster training means faster experimentation / agent iterations etc.
>
>
> **"An agent designer has to ...". Can you elaborate a bit more on the setting? Ideally, describe this trade-off in more detail. Sometimes, sample-efficiency and time are intertwined, if your data collection is slow, then sample inefficiency will bump the wall-clock time.**
>
> Our wording here was meant to highlight that even for non-strictly-latency-constrained areas like distributed RL, choosing improved sample efficiency does not necessarily result in better runtime performance. This means that for practical applications, an agent designer might prefer LSTMs over transformers due to shorter wall-clock training times. However given reviewer comments, our focus on distributed RL in this wording might obscure our original motivation, the solution of actor-latency-constrained settings with distributed RL as a testbed. In the next revision, we would like to position the paper more clearly as working to solve the actor-latency-contrained setting, with a particular application to training transformers for RL, and will rephrase this paragraph.

---

> ### Author Response · Authors · 2020-11-18
> **Re: Review Part 2**
>
> **Plot the curves till convergence of all the algorithms?**
>
> Figure 1 was mainly meant as a motivating figure, so we ran both models long enough to show the inversion of the results when changing x-axis from environment steps to wall-clock time.
> For the other figures, we had set a specific number of env steps as a threshold to terminating the experiment as is typically done in RL. The specific env step threshold was chosen during very early experiments where we found that an (un-hyper-optimized) GTrXL could solve the task. Alternatively we could have chosen a wall-clock time threshold to terminate the experiments. We found this was undesirable for several reasons. The main reason being that it requires each run, even ones with very fast / low complexity models, to run for as long as the slowest model. Since we need to run each model serially on the same reference machine, this would have significantly increased the amount of compute necessary to obtain results.
> We include the untruncated graphs in the appendix. Note that the x-scale can sometimes make it difficult to see the salient aspects of the relative performance of each model, as the GTrXL is often run for much longer in terms of wall-clock. The truncation on wall-clock time was chosen specifically to highlight the relative performance improvements of the ALD framework over other baselines.
>
>
> **For the table on top of the page 3, how do you get the numbers?**
> The table in Figure 1 is a measurement of (1) how much time the learner process of the respective models spent waiting for data from actors and (2) the environment steps per second generated on a single actor process. For (1), we ran each model for 10 minutes and took the average over the 10 minutes that the learner process spent waiting for a buffer of data versus doing anything else. For (2), for each model we counted how many steps were completed in 5 seconds and then divided by 5. We averaged this over 50 five second increments.
>
>
> **Why did you decide to do you present the per-seed curves on 15x15 maze while presenting the aggregated results for 9x9? How do these look on 9x9 maze?**
>
> The reason to present the per-seed curves on 15x15 was that there was a noticeable effect where the LSTMs got stuck in a local maximum where they would enter any goal without discernment of the indicator color. This did not happen as significantly for the ALD LSTM, and this is what we wanted to demonstrate with the per-seed curves here. Since 9x9 is shorter-horizon, we did not see such a noticeable effect there. For completeness we include the per-seed curves of 9x9 in the appendix.
>
>
> **Other comments**
>
> Thank you for these valuable corrections to our text. We will apply these suggestions in the next revision.

---

> > ### Comment · AnonReviewer2 · 2020-11-20
> > **Response to response**
> >
> > Thanks for your reply! I think the authors did a good job of clarifying and repositioning the paper. I do not have any more questions to the authors. I tend to raise the score, but I want to have a discussion with other reviewers/AC first.

---

### Official Review · AnonReviewer3 · 2020-10-28
**Actor-learner distillation for distributed RL**

**Rating:** 7
**Confidence:** 4

**Review:**

##########################################################################

Summary:

The paper proposes a solution to actor-latency constrained settings in RL by using policy distillation to compress a large “learner model” towards a more tractable “actor model”. In particular, it proposes to exploit the superior sample efficiency of transformer models while utilising an LSTM-based actor during execution. The proposed procedure, called Actor-Learner Distillation (ALD), provides comparable performance to transformers in terms of sample efficiency, yet produces a wall-clock run-time that's on par with LSTM agents.

##########################################################################

Reasons for score:

I'm voting for accepting the paper. I like the idea of policy distillation between transformer-based learner and LSTM-based actor in the distributed RL framework. The paper is written with great clarity and is easy to follow. I don't have major concerns regarding this work but hope that the authors can address my minor concerns in the rebuttal period.

##########################################################################

Pros:

1. The work addresses an important problem for distributed RL: whether it is possible to utilise large model capacity while still acting within the computational limits imposed by the problem setting and do all of this during the training. The problem itself is real and practical, e.g., in robotic operations with limited computation resources.
2. The process of policy distillation between transformer-based learner and LSTM-based actors is described in great detail and provides additional insight into concrete decisions made for the method, such as the value distillation loss for the actor model and Distil/RL step ratio.
3. This manuscript includes thorough experimental of the proposed method that includes both qualitative analysis and quantitative results. The choices regarding the two environment tasks and baselines are reasonable.

##########################################################################

Cons:

1. Regarding the DpRL hyperparameter (actor SPS / learner SPS), it isn't very clear to me how this ratio has been manipulated for instance in Figure 3. Is this achieved using modifications of actor/learner size? I'd appreciate it if the authors could add clarification on this.
2. On the result figures where the X-axis corresponds to Wall-Clock Time, it isn't clear to me what was used as a basis for choosing the maximum range of the X-axis. For example, Figure 4 (Right) was cut after 5 minutes, where ALD is much higher than the baselines. However, I'd be interested to know when do ALDs and GTrXL (which achieves the same result in eventually) meet on the diagram.

##########################################################################

Questions during rebuttal period:

Please address and clarify the cons above

---

> ### Author Response · Authors · 2020-11-18
> **Re: Actor-learner distillation for distributed RL**
>
> Dear Reviewer,
>
> Thank you for your review and positive score. We hope to clarify any concerns you had below:
>
> **DpRL hyperparameter (distill SPS / rl SPS)**
> For the case of Figure 3, the ratio is being manipulated through a hard constraint: the main thread estimates the learner and distill SPS, and then causes either learner or distill threads to sleep a fraction of the time to obtain the target SPS ratio. This strict bottleneck was used to artificially experiment with the sensitivity of ALD to the DpRL ratio. In other experiments, we do not do any strict bottleneck but try to improve the distillation speed through the use of HOGWILD! parallelization.
>
> **On the result figures where the X-axis corresponds to Wall-Clock Time, what was used as a basis for choosing the maximum range of the X-axis**
> For all models, we had set a specific number of env steps as a threshold to terminating the experiment, which is the standard in RL experiments. The specific env step threshold was chosen during very early experiments where we found that an (un-hyper-optimized) gated transformer could solve the task. Alternatively we could have chosen a wall-clock time threshold to terminate the experiments. We found this was undesirable for several reasons. The main reason being that it requires each run, even ones with very fast / low complexity models, to run for a specific amount of time. Since to make sure our timing results make sense, we need to run all experiments serially on the same reference machine, which means now we need to run all models + all seeds for as much time as it takes for the GTrXL to do the desired env steps.
>
> We include the untruncated graphs in the updated appendix. Note however that the x-scale can sometimes make it difficult to see the salient aspects of the relative performance of each model, as the gated transformer (GTrXL) is often run for much longer in terms of wall-clock. The truncation on wall-clock time was chosen specifically to highlight the relative performance improvements of the ALD framework over other baselines.

---

> > ### Comment · AnonReviewer3 · 2020-11-18
> > **Thanks for your reply**
> >
> > Dear Authors,
> >
> > Thank you for answering my questions.
> >
> > I also appreciate you adding the untruncated wall-clock time graphs and HOGWILD! summary in the Appendix.

---

### Official Review · AnonReviewer4 · 2020-10-28
**A well written paper exploring a simple but sensible idea**

**Rating:** 8
**Confidence:** 4

**Review:**

The main aim of this paper is to increase the efficiency (in terms of wall clock time) of Transformer based models for reinforcement learning.

Previous works have shown Transformer-like models to be highly performant across a range of domains, including recent results on reinforcement learning (with gated Transformer-XLs). However a drawback of these models  (over say, LSTMs) is their relatively slower inference speeds. This is especially a problem in RL settings where Actors are typically run on CPUs (not GPUs) and send trajectories to a central learner. The problem here is two-fold: 1) slower overall training time due to high latency with the learner waiting on actors 2) slower inference post training in latency sensitive deployment settings like robotics/other control based settings.

The solution proposed here (the method is referred to as "Actor-learner distillation") is to instead use LSTMs for acting and Transformers for learning. Typically the learner would send parameters to the actors every update -- as this is not possible in this hybrid approach, a distillation loss is instead suggested as a means of updating the actors (with a replay buffer). The authors successfully show that this transfers both "good policies" and the relevant inductive biases from the Transformer as well allowing fast inference as expected. Further, the "off policy"-ness of the model does not turn out to be an issue in the domains considered. The results are promising and the the writing very clear. A few points below:

* One question I have is whether the authors ran an experiment replacing the LSTMs in the actors simply with smaller transformer models? For example, 2 layers instead of 4 or smaller embedding sizes? This would also allow faster inference while inherently retaining some of the inductive biases.

* As a second baseline: while this wouldn't solve the issue of slower training, one could achieve faster inference of the trained model simply by distilling the final Transformer into an LSTM (and then optionally fine-tuning). Are the authors able to report numbers for these?

* Further, in between the two extremes of only LSTMs acting or only Transformers one could also consider more hybrid approaches that start with a Transformer to learn a very good starting policy in a few steps and then distill this into an LSTM online similar to work here (https://arxiv.org/pdf/1806.01780.pdf). The worked linked here performs "online distillation" combined with a convex combination of model outputs which would then negate the need for a replay buffer (though would require running both models on the actor until fully switching over to the LSTM).

* While the results are conclusive, the work could be improved by running on larger (or 3-dimensional) environments as considered in the original Gated Transformer paper this builds on.

* Lastly, are the authors able to provide possible future directions to be considered that could boost this work further?

Overall this is well written paper exploring a conceptually simple but well-implemented and important idea. As models grow in size, techniques like this will be very important to allow scaling up of RL capabilities. I would recommend accepting this paper.

---

> ### Author Response · Authors · 2020-11-18
> **Re: A well written paper exploring a simple but sensible idea**
>
>
> Dear Reviewer,
>
> Thank you for your review and positive score. We hope to answer your questions below:
>
> **An experiment replacing the LSTMs in the actors with small transformer models**
> We did try a preliminary run with smaller transformers as the actors. However we found that even if it had fewer layers, the attention operations still required a significant amount of time when running on CPU. We could not reduce the memory window size of the smaller transformer as this might potentially reduce the receptive field such that it would not be possible to solve the task. Therefore for future work, we hope to explore potentially integrating ALD where actors are using batched inference on a collective accelerator, and this would allow larger models to be used during acting while still maintaining a target latency.
>
> **Distilling the final Transformer into an LSTM + hybrid approaches that start with a Transformer and then distill into an LSTM online**
> We think the standard model compression procedure, where a large model is first trained and then distilled into smaller ones, is a powerful tool that will continue to be applicable in many areas. However we would like to highlight a significant difference between ALD and standard model compression, which is we explicitly want to consider latency-constrained settings where the more expressive policy cannot be run reliably during acting. This presents unique challenges not generally present in standard distillation / model compression settings, such as an inherently off-policy but online learning setting for the teacher model. For baselines, we only wanted to compare against methods which similarly maintained this actor-latency constraint, and the nearest method we found in the literature is the “asymmetric-actor-critic” method we presented in our results. The gated transformer curves we show are only meant as a point of comparison as the target sample efficiency we are aiming for with ALD. We will make this motivation clearer in the next revision.
> With respect to Mix&Match, we think this is a very relevant work and was one that we were originally aiming to cite as a reference. Thank you for bringing it to our attention.
>
>
> **Running on larger 3-dimensional environments as in the Gated Transformer paper**
> We agree that running on dmlab environments would enable us to test the effectiveness of ALD at large scale.  Unfortunately there was some significant difficulty in getting a working model at the compute scale we had available. One problem was that, at the smaller transformer size (4 layers w/ 64 memlen in our case v.s. 12 layers w/ 512 memlen for the original result), the CNN to process the 96x72x3 image became a more significant bottleneck on CPU. Additionally, with the 4 layer+64 mem gated transformer and a smaller CNN we were not able to solve psychlab_arbitrary_visuomotor_mapping in the 1B env steps they reported for their model in the paper. This model took over 16 days to train to 1B steps and thus we were not able to experiment with a varied set of hyperparameters in the interim since submitting.
>
> **Possible future directions**
> Yes, we added a discussion of potential future directions to the updated paper draft. We repost it here for convenience:
> In future work, we wish to investigate whether integrating the ALD procedure with batched inference for actors would still maintain the same performance increases we demonstrated in our results, while at the same time enabling larger actor models to be used and correspondingly larger learners.

---

> > ### Comment · AnonReviewer4 · 2020-11-20
> > **Re**
> >
> > Thanks for your response, that answers all the main questions I had.

---

### Author Response · Authors · 2020-11-18
**Rebuttal Revision**

Dear Reviewers,

We have incorporated your valuable feedback into a new paper revision. To summarize the main changes:
  - A re-written introduction section that more clearly positions the paper and centers its motivation around the actor-latency-constrained setting.
  - Minor edits and typo corrections pointed out by reviewers.
  - An expanded background briefly covering V-MPO and IMPALA before the method section.
  - Full wall-clock curves in the appendix for the graphs originally shown in figures 4 and 5.
  - Per-seed curves for 9x9 I-Maze.
  - A summary of HOGWILD! in the appendix.
  - A correction stating that ALD is using V-Trace corrections. The correction was due to a bug where the boolean argument passed was unintentionally inverted.

---

> ### Comment · AnonReviewer2 · 2020-11-19
> **Can't find the revision pdf**
>
> Hi! Thanks for your updates, but have you uploaded the new version of the paper? I can't find it in the 'revisions' tab.
>
> UPD: Weird, it's there after a refresh.

---

### Decision · Program_Chairs · 2021-01-07
**Final Decision**

**Decision:**

Accept (Poster)

**Comment:**

I thank the authors for their submission and participation in the author response period. The reviewers unanimously agree that the papers proposes an interesting and original approach to using a costly model on a learner node, while distilling to a cheaper model run on actor nodes to gather experiences in a distributed RL framework. During discussion, R1 and myself emphasized the concern that the experiments in this paper leave open the question whether the approach will work beyond toy environments. However, I side with R2 and R3 in that the paper presents a valuable contribution to the community as it stands, and that the experiments proof the concept to the point that the paper should be accepted. I therefore recommend acceptance.